# Lightweight Single-Stage Ship Object Detection Algorithm for Unmanned Surface Vessels Based on Improved YOLOv5

**DOI:** 10.3390/s24175603

**Published:** 2024-08-29

**Authors:** Hui Sun, Weizhe Zhang, Shu Yang, Hongbo Wang

**Affiliations:** State Key Laboratory on Integrated Optoelectronics, College of Electronic Science and Engineering, Jilin University, Changchun 130012, China; sunhui22@mails.jlu.edu.cn (H.S.); zhangwz23@mails.jlu.edu.cn (W.Z.); shuyang1922@mails.jlu.edu.cn (S.Y.)

**Keywords:** object detection, attention mechanism, YOLO, unmanned surface vehicle, lightweight detection algorithm

## Abstract

Object detection is applied extensively in various domains, including industrial manufacturing, road traffic management, warehousing and logistics, and healthcare. In ship object detection tasks, detection networks are frequently deployed on devices with limited computational resources, e.g., unmanned surface vessels. This creates a need to balance accuracy with a low parameter count and low computational load. This paper proposes an improved object detection network based on YOLOv5. To reduce the model parameter count and computational load, we utilize an enhanced ShuffleNetV2 network as the backbone. In addition, a split-DLKA module is devised and implemented in the small object detection layer to improve detection accuracy. Finally, we introduce the WIOUv3 loss function to minimize the impact of low-quality samples on the model. Experiments conducted on the SeaShips dataset demonstrate that the proposed method reduces parameters by 71% and computational load by 58% compared to YOLOv5s. In addition, the proposed method increases the mAP@0.5 and mAP@0.5:0.95 values by 3.9% and 3.3%, respectively. Thus, the proposed method exhibits excellent performance in both real-time processing and accuracy.

## 1. Introduction

Advances in 5G communication and artificial intelligence have advanced the development of unmanned surface vessels considerably. Such vessels play a crucial role in various fields, e.g., hydrological monitoring and maritime search and rescue, due to their low cost, low power consumption, and intelligent features. Key technologies for unmanned surface vessels include perception, control, and communication [1]. During mission execution, perception technology provides unmanned surface vessels with essential environmental information. For example, ship object detection technology enables unmanned surface vessels to identify surrounding ships, thereby offering vital positional and ship type information for effective collision avoidance and route planning. Thus, ship object detection is a fundamental component of unmanned vessel technology.

In ship target detection tasks, commonly used sensors include LiDAR and cameras. Note that the mainstream detection range of LiDAR is 150 m, however, in a maritime environment, ships are typically dispersed widely to ensure safe navigation, thereby resulting in sparse point clouds at greater distances. This sparsity causes LiDAR to provide insufficient information for target recognition, and processing LiDAR point clouds requires powerful hardware. In contrast, cameras have a larger detectable distance and produce visible light images with higher pixel density and rich texture information. Thus, cameras offer more data for target detection on unmanned ships. Studying target detection algorithms for visible ship images is crucial to ensure the safe navigation of unmanned vessels [2].

Object detection algorithms can be categorized into traditional methods and deep learning-based methods. Traditional methods primarily rely on handcrafted feature construction. In 2005, Dalal and Triggs proposed the histogram of oriented gradients (HOGs) feature descriptor [3]. To detect objects of various sizes, the HOG detector maintains a constant detection window size while performing multiple scales on the input image. In addition, DPM [4], which is an extension of HOG features, views training as a method for learning to decompose an object, with inference seen as a collection of detections of different object parts. Due to the limitations of handcrafted features, traditional methods often suffer from poor robustness against environmental changes, which leads to false positives and missed detections.

Deep learning-based object detection algorithms have advanced due to the strong feature learning capabilities of convolutional neural networks (CNNs). These algorithms are primarily divided into single-stage and two-stage object detection algorithms. Two-stage detection involves two steps. First, candidate regions are generated, and then algorithms, e.g., R-CNN [5], Fast R-CNN [6], and Faster R-CNN [7], are employed to classify and regress the candidate regions. In contrast, single-stage object detection networks, e.g., YOLO [8,9,10] and SSD [11], employ CNNs to extract the feature information of the targets and then sample and classify/regress the corresponding feature maps using anchor boxes of different aspect ratios. Note that two-stage methods offer higher accuracy, however, they struggle to satisfy real-time requirements. In contrast, single-stage object detection networks have an advantage in terms of real-time performance due to their fast inference speed, thereby making them suitable for rapid detection in maritime environments for unmanned surface vessels. In addition, single-stage object detection networks are being improved continually to balance accuracy and real-time performance, including advancements like YOLOv7 [12] and YOLOv10 [13].

Given that unmanned surface vessels typically have limited computational power, lightweight models are essential. Thus, this paper investigates the lightweight model of single-stage networks using YOLOv5s from the YOLO series as a baseline to design an efficient object detection network. The reasoning is that after extensive practice and optimization, YOLOv5 has proven its reliability across a wide range of scenarios. Thanks to its prolonged development period, the YOLOv5 model is relatively stable, with fewer potential issues. As the smaller variant in the YOLOv5 series, the YOLOv5s strikes an excellent balance between performance and efficiency. It performs well in resource-constrained environments, maintaining high detection accuracy while ensuring faster inference speeds. The primary contributions of this study are summarized as follows:To address the high number of parameters in the backbone of the YOLOv5s network, we introduce an improved ShuffleNetV2 network as the backbone. This modification attempts to maintain the model’s feature extraction capability while reducing the parameter count.We designed a split-DLKA attention module that leverages the ability of large kernels to expand the receptive field and the capacity of deformable convolutions to adjust to the convolution kernel shape adaptively. This enhances the network’s adaptability to samples of different shapes, thereby making it suitable for detecting vessels of varying sizes in maritime scenarios.By incorporating the WIOUv3 loss function into the network, the impact of low-quality samples on the model is reduced, which results in improved detection accuracy.

## 2. Related Work

Ship detection is a specialized area within object detection that requires balancing two key objectives: accuracy and real-time performance. Given the significant variation in ship sizes, the network must also have strong multi-scale detection capabilities. One of the most effective ways to enhance a neural network’s object detection accuracy is by improving its feature extraction capabilities, often achieved by increasing the network’s depth. However, this approach also increases computational complexity and the number of parameters, which can negatively impact the network’s real-time performance. Consequently, numerous scholars have conducted extensive research on improving accuracy, real-time efficiency, and the multi-scale detection capabilities of networks.

### 2.1. Attention Mechanism in Ship Detection

The attention mechanism plays a crucial role in enhancing feature extraction and multi-scale detection capabilities. For instance, Shen et al. [14] introduced a multiple information perception-based attention module (MIPAM). Their approach incorporates channel-level information collection through global covariance pooling and channel-wise global average pooling, while spatial-level information is collected in a similar way. This method enriches feature representation, leading to improved detection accuracy when integrated into the YOLO detector. Due to the diverse shapes of ships and the complexities of environmental interference, multi-scale detection has become an essential capability for ship detection networks. Many experts utilize attention mechanisms to dynamically adjust the weights within a network, designing and implementing them to enhance the network’s ability to capture important information across different scales. In [15], the author proposed an attention feature filter module (AFFM), which uses attention supervision generated from high-level semantic features in the feature pyramid to highlight information-rich targets, forming a spatial attention mechanism. Unlike traditional attention mechanisms such as CBAM, the attention signals in the AFFM are derived from higher-level feature maps, which better represent the distinctive characteristics of nearshore ships. Guo et al. [16] utilized sub-pixel convolution, sparse self-attention mechanisms, channel attention, and spatial attention mechanisms to enhance semantic features layer-by-layer, which ensures that the feature map contains richer high-level and low-level semantic information, effectively improving the detection performance of small ships. Yao et al. [17] designed a feature enhancement module based on channel attention, increasing the emphasis on ship features and expanding the receptive field through an adaptive fusion strategy. This enhances spatial perception for ships of varying sizes. Li et al. [18] developed an adaptive spatial channel attention module, effectively reducing the interference of dynamic background noise on large ships. Additionally, they designed a boundary box regression module with gradient thinning, improving gradient sensitivity and multi-scale detection accuracy. Li et al. [19] took a more intuitive approach by incorporating a transformer into the YOLOv5 backbone to enhance feature extraction. Zheng et al. [20] integrated a local attention module into a SSD to improve the detection accuracy of smaller models. Li et al. [21] enhanced YOLOv3 by adding the CBAM attention mechanism to the backbone, enabling the model to focus more on the target and thereby improving detection accuracy.

### 2.2. Attention Mechanism Combined with Other Improvements

Some authors combine attention mechanisms with other techniques, such as improved loss functions and enhanced convolution methods, to not only boost feature extraction capabilities but also reduce network parameters and improve real-time performance. For example, Zhao et al. [22] introduced the SA attention mechanism into YOLOv5n to enhance feature extraction and replaced standard convolution in the neck with Ghost Conv, reducing both network complexity and computational load. In [23], the author proposed a lightweight LWBackbone to decrease the number of model parameters and introduced a hybrid domain attention mechanism, which effectively suppressed complex land background interference and highlighted target areas, achieving a balance between precision and speed. Ye et al. [24] incorporated the CBAM attention module into the backbone of YOLOv4, replaced CIOU with EIOU, and substituted non-maximum suppression (NMS) with soft-NMS to reduce missed detections of overlapping ships. Bowen Xing et al. [25] added the CBAM attention mechanism to FasterNet, replaced the YOLOv8 backbone, and introduced lightweight GSConv convolution in the neck to enhance feature extraction and fusion. Additionally, they improved the loss function based on ship characteristics and integrated MPDIoU into the network, making it more suitable for ship detection.

### 2.3. Data Enhancement Prevent Overfitting

Some experts and scholars focus on the ship data themselves, utilizing data enhancement and preprocessing as primary methods to improve network detection performance, along with other enhancements. For instance, Zhang et al. [26] developed a new data enhancement algorithm called Sparse Target Mosaic, which improves training samples. By incorporating a feature fusion module based on attention mechanisms and refining the loss function, they were able to enhance detection accuracy. Gao et al. [27] applied gamma transform to preprocess infrared images, increasing the gray contrast between the target and background. They also replaced the YOLOv5 backbone with MobileNetV3, reducing parameters by 83% without significantly compromising detection performance. Chen et al. [28] designed pixel-space data enhancement in a two-stage target detection network, using set transformation and pixel transformation to enhance data diversity and reduce model overfitting. This approach improved network focus and accuracy, achieving a remarkable mAP of 99.63%. Qiu et al. [29] addressed the singleness of the dataset’s image style in ship detection datasets by proposing an anti-attention module. They inputted the original feature layer into a trained convolutional neural network, filtered the output weights, and removed feature layers that negatively impacted detection. This led to improvements in both the mAP and F1-score.

### 2.4. Improvement in Lightweight

Some scholars focused on improving convolutional methods, backbone, and loss functions to enhance ship detection performance. For example, Li et al. [30] integrated OD-Conv into the YOLOv7 backbone, effectively addressing the issue of complex background interference in ship images, thereby improving the model’s feature extraction capabilities. Additionally, they introduced the space-to-depth structure in the head network to tackle the challenge of detecting small- and medium-sized ship targets. These improvements led to a 2.3% increase in mAP compared to the baseline. Zheng et al. [31] proposed a differential-evolution-based K-means clustering method to generate anchors tailored to ship sizes. They also enhanced the loss function by incorporating focal loss and EIOU, resulting in a 7.1% improvement in average precision compared to YOLOv5s. Shi et al. [32] introduced the theta-EIOU loss function, which enhances the network’s learning and representation capabilities by reconstructing the bounding box regression loss function, improving background partitioning, and refining sample partitioning. The improved method outperformed the original YOLOX network. Zhang et al. [33] incorporated a multi-scale residual module into YOLOv7-Tiny and designed a lightweight feature extraction module. This reduced the number of parameters and computational load of the backbone while improving feature extraction accuracy. They used Mish and SiLU activation functions in the feature extraction module to enhance network performance and introduced CoordConv in the neck network to reduce feature loss and more accurately capture spatial information. Zheng et al. [34] replaced YOLOv5s’ original feature extraction backbone with the lightweight MobileNetV3-Small network and reduced the depth-separable convolutional channels in the C3 module to create a more efficient feature fusion module. As a result, the final network had 6.98MB fewer parameters and an improved mAP compared to YOLOv5s.

Due to the significant variations in the appearance and morphology of ships, scale distortion can affect detection accuracy. In addition, buildings along inland waterways can impact ship detection. Thus, enhancing the model’s robustness to size variations is essential. For deployment on unmanned surface vessels, reducing both the number of model parameters and the computational load while maintaining sufficient detection accuracy is crucial. The goal of this study is to strike a balance between these goals and improve the detection and differentiation of various ship types.

## 3. Introduction to YOLOv5s Algorithm

The structure of the YOLOv5 model is shown in Figure 1. YOLOv5 is widely used in engineering practices and comes in four versions: YOLOv5s, YOLOv5m, YOLOv5l, and YOLOv5x. These versions have progressively increased model width, depth, and parameter count, which results in improved detection accuracy. YOLOv5s is the smallest and fastest version, making it suitable for deployment on devices with low computing power. The model comprises four main components: input, backbone, neck, and prediction layers (head). The characteristics of each part are summarized as follows:-Input: The input layer processes raw images, resizing them to a standard dimension and normalizing the pixel values. This process prepares the data for further processing by the model.-Backbone: YOLOv5′s backbone is based on the cross-stage partial network (CSPNet) architecture. The main purpose of the backbone in YOLOv5 is to extract features from the input image through a series of convolutional layers. CSPNet helps enhance the model’s learning capability and reduces computational complexity by partitioning the feature map of the base layer into two parts and then merging them through a cross-stage hierarchy.-Neck: The neck network of YOLOv5 employs the path aggregation network (PANet) as its feature fusion module. PANet enhances feature fusion through both top–down and bottom–up pathways, which enables the network to fully leverage feature information from different levels. This design ensures effective utilization of multi-scale features, which improves the detection of objects of various sizes and shapes.-Head: The head of YOLOv5 contains three detection heads, which can predict objects at multiple scales simultaneously, thereby improving both accuracy and efficiency. Note that YOLOv5 uses CIOU loss as the bounding box loss function and weighted non-maximum suppression (NMS). CIOU loss improves bounding box regression by considering the overlap area, center point distance, and aspect ratio, and the weighted NMS increases the suppression weight of high-confidence bounding boxes, thereby helping to obtain more reliable results.

## 4. Proposed Method

Considering the power consumption issues under a limited power supply, the hardware of unmanned surface vessels typically has low power consumption, which in turn results in highly limited computational power. In addition, the memory of unmanned surface vessels is often limited. Thus, it is crucial to reduce the computational load and memory usage of the object detection network. In YOLOv5s, the backbone accounts for 66% of the total FLOPs (floating-point operations). To achieve a lightweight network, the computational load of the backbone must be reduced. Further, the role of the attention mechanism in terms of enhancing the local features and integrating global information can mitigate the feature extraction deficiencies caused by a lightweight backbone network.

The structure of the proposed lightweight object detection network is shown in Figure 2. The model improvements focus on the backbone, neck, and loss function.

### 4.1. Backbone

#### 4.1.1. ShufflenetV2 Backbone

Each stage of the ShuffleNetV2 [35] backbone comprises multiple blocks. The structural parameters of ShuffleNetV2 are detailed in Table 1.

In each stage, the first block is a down sampling module, and the remaining three blocks are non-down sampling modules.

#### 4.1.2. Activation Function

In ShuffleNetV2, the activation function is ReLU, which is defined as follows:(1)$$f(x)=\max\left(0,x\right)$$

ReLU is a single-sided saturation function with robustness against noise interference. ReLU truncates negative values to zero, thereby introducing sparsity and improving computational efficiency (Figure 3a). However, the ReLU activation function outputs zero for negative input values, which can lead to neurons becoming “dead” or inactive during training, meaning their gradients are consistently zero and they cannot contribute to learning. The Leaky ReLU function addresses this issue by allowing a small, non-zero slope in the negative region, ensuring that all neurons remain partially active and continue to update during training. By preserving a small gradient for negative inputs, Leaky ReLU helps maintain gradient flow throughout the network, which enhances both the stability and convergence speed of the training process. This enhancement is particularly crucial for our object detection tasks, as these models often require deep network architectures to effectively capture complex features. The formula for LeakyReLU is given as follows:(2)$$f(x)=\max\left(0,x\right)+\xi \cdot \min\left(0,x\right)$$
where the parameter ξ is generally set to 0.01. In the part where the input is less than zero, the function value is a negative value with a small gradient (rather than zero), as shown in Figure 3b.

When the input to the activation function is less than zero, the gradient can still be calculated. In our experiments, we observed that Leaky ReLU outperforms ReLU in terms of performance. Specifically, the use of Leaky ReLU resulted in the improved detection accuracy of the model. Experimental results show that in the ship object detection task, replacing the activation function of ShuffleNetV2 with LeakyReLU and using it as the backbone of YOLOv5s can yield a 0.3% increase in mAP@0.5 and a 7.7% increase in precision.

### 4.2. Attention Split-DLKA Module

In the task of ship target detection, the size and shape of different ship types can vary significantly, encompassing a wide range of morphologies and structures. This diversity necessitates that the detector be capable of adapting to multiple forms, which increases model complexity and makes it challenging to generalize across different ship types. Additionally, the maritime environment, including oceans, harbors, and other water contexts, is often complex and dynamic, with potential disturbances such as waves, other vessels, and infrastructure. Moreover, the diversity in ship sizes and shapes can lead to sample imbalance in the training data, resulting in sub-optimal model performance for certain ship categories.

An effective strategy to address these challenges is to expand the network’s receptive field, which defines the input region size on which the features at a particular network position depend. The receptive field is calculated as follows:(3)$$r_{l}=r_{l-1}+\left(k_{l}-1\right)\cdot j_{l-1}$$
where $r_{l-1}$ represents the receptive field size corresponding to layer l−1, $k_{l}$ denotes the kernel size or pooling size of layer l, and $j_{l-1}$ represents the pixel distance between adjacent elements on the feature map. From Equation (3), it is evident that increasing the kernel size can result in a larger receptive field, ultimately producing many equivalent feature scales; a larger receptive field allows the model to capture a broader range of contextual information, which is crucial for detecting objects of different scales and shapes.

The structure of split-DLKA is illustrated in Figure 4. A tensor x of size [B,C,H,W] is divided into i subsets along the channel dimension, denoted as $x_{i},\;\;i=1,\;\;2\ldots$, where i=2. Here, each subset $x_{i}$ has $\frac{C}{2}$ input channels, which are passed separately to the deformable DW module.

To expand the receptive field of the neural network, large convolutional kernels of size 5 × 5 and 7 × 7 are applied to the deformable DW module, which can adaptively adjust its sampling positions to better align with the local features of the input data, thereby enhancing feature extraction capability. This improvement allows the model to adapt more effectively to geometric transformations, enabling the network to better handle irregularly shaped objects or features.

In the deformable DW module, 2D convolution can be described as using a grid to sample the feature map and the sampled values are assigned weights and summed; an offset is generated by convolving the input feature map. This offset $\Delta p_{n}$ (which is typically a non-integer offset) is added to the original sampling positions to obtain the new sampling positions. This process is expressed as follows:(4)$$y\left(p_{0}\right)=\sum_{p_{n}\in R}w\left(p_{n}\right)\cdot x\left(p_{0}+p_{n}+\Delta p_{n}\right)$$where x is feature map, y is output feature map, $p_{n}$ is the offset of each point in the convolution kernel relative to the center point $p_{0}$ and, $w\left(p_{n}\right)$ is the weight of each deviation.

Bilinear interpolation is employed to obtain the pixel values at the offset positions, and the expression is as follows:(5)$$x\left(p\right)=\sum_{q}G\left(q,p\right)\cdot x\left(q\right)$$
where p is $p_{0}+p_{n}+\delta p_{n}$ in Equation (4), q is the coordinate of an integer point, and $G\left(q,p\right)$ is the bilinear interpolation kernel function.
(6)$$G\left(q,p\right)=g\left(q_{x},p_{x}\right)\cdot g\left(q_{y},p_{y}\right)\cdot x\left(q\right)$$
here, $g\left(a,b\right)=\max\left(0,1-\left|a-b\right|\;\right)$. This restricts the interpolation point and its neighboring points to be within a distance of no greater than one pixel.

The parameter calculation for the split-DLKA module is described as follows:

The convolution generates an offset with an output channel number of 2⋅k⋅k. The parameter count for this process is $2C_{in}k^{4}+2k^{2}$.

Then, a deformable convolution is performed using the convolution and previously generated offset with the groups set to $C_{in}$. The parameter count for this process is $C_{in}k^{2}$.

The parameter count of the split-DLKA is as follows: (7)$$P(\mathrm{split}\text{-}\mathrm{DLKA})=(C_{in}+2)k^{2}+2C_{in}k^{4}$$

### 4.3. Loss Function

In YOLOv5, the complete intersection over union (CIOU) is used for bounding box regression. The CIOU considers the distance between the center points of the ground truth and the predicted box, as well as the aspect ratio consistency, as shown in Equation (8):(8)$$loss_{CIoU}=1-IoU+\frac{\rho^{2}\left(b^{pre},b^{gt}\right)}{c^{2}}+\chi v$$
where IoU is the ratio of the area of overlap between the predicted bounding box and the ground truth bounding box to the area of their combined region, $\rho \left(b^{pre},b^{gt}\right)$ is the distance between the center point of the predicted box and the real box, c is the diagonal distance of the smallest external rectangle, $v=\frac{4}{\pi^{2}}{\left(\arctan\frac{w^{gt}}{h^{gt}}-\arctan\frac{w^{pre}}{h^{pre}}\right)}^{2}$ measures the aspect ratio between the predicted box and the ground truth, and $\chi =\frac{v}{\left(1-IoU\right)+v}$ is the balance parameter, which increases when the predicted and target boxes increase in size. 

When the training data contain low-quality examples, the geometric factors (distance and aspect ratio) increase the penalty, thereby reducing the model’s generalizability. To address this issue, WIOUv1 [36] introduces a distance attention mechanism that can amplify the ${loss}_{IOU}$ of the ordinary-quality anchor box considerably.

(9)$$loss_{WIOUv1}=R_{WIOU}\cdot loss_{IOU}$$(10)$$R_{WIOU}=exp\left(\frac{{\left(w_{2}-w_{1}\right)}^{2}+{\left(h_{2}-h_{1}\right)}^{2}}{\left(w^{2}+h^{2}\right)}\right)$$
where $w_{1}$, $h_{1}$ is the coordinate of the ground truth, $w_{2}$, $h_{2}$ is the coordinate of the predicted box, and w, h are the width and height of the smallest enclosing box, respectively (Figure 5). Inspired by focal loss [37], a monotonic focusing coefficient $L_{IoU}^{\gamma *}$ is constructed to reduce the contribution of easy examples to the loss value and obtain WIOUv2.
(11)$$loss_{WIOUv2}=L_{IoU}^{\gamma *}\cdot loss_{WIOUv1},\gamma >0$$

To prevent large harmful gradients from low-quality examples, a non-monotonic focusing coefficient, which is also the gradient gain $r=\frac{\beta }{\delta \alpha^{\beta -\delta }}$, is constructed, where β is the outlier degree of the anchor box, and α, δ are dynamic parameters.
(12)$$loss_{WIOUv3}=r\cdot loss_{WIOUv1}$$

The WIOUv3 loss function replaces the original loss function, thereby reducing the model’s overfitting and enhancing its detection performance. Additionally, assigning a small gradient gain to the anchor box with a large outlier degree will effectively prevent large harmful gradients from low-quality examples. It can be seen from Figure 6 that when α=1.9,δ=3, we can obtain the biggest gradient and the smallest outlier degree. 

## 5. Experiment

Multiple experiments were conducted to demonstrate the effectiveness of the proposed method. First, a comparative experiment was performed with existing single-stage object detection networks to validate the advantages of the proposed method in terms of its real-time performance and accuracy. Then, ablation experiments were conducted to verify the effectiveness of the proposed split-DLKA module and the other improvements implemented in the proposed method. Finally, comparative experiments were conducted with other attention mechanisms to validate the effectiveness of split-DLKA in terms of improving mAP.

### 5.1. Experimental Dataset

The dataset used in this study was the SeaShips dataset [38], which contains 7000 images (resolution: 1920 × 1080). This dataset includes six common types of ships, i.e., ore carriers, container ships, bulk carriers, general cargo ships, fishing boats, and passenger ships. The distribution of the number of each type of ship is shown in Figure 7, examples of various ship data are shown in the Figure 8.

The dataset was divided into training, validation, and test sets at a ratio of 6:2:2. In addition, data augmentation techniques, e.g., cropping and stitching, were applied to enhance the dataset. The augmented data are illustrated in Figure 9.

### 5.2. Experimental Platform and Parameter Settings

The experiments conducted in this study were conducted with Ubuntu 20.04 system with a 12th Gen Intel(R) Core (TM) i7-12700F CPU (sourced from Intel, Santa Clara, CA, USA) with 16 GB of RAM and an NVIDIA GeForce RTX 3060 GPU (CUDA version: 12.1). The specific details are shown in Table 2.

In this experiment, training was performed over 100 epochs with a learning rate of 0.01, momentum of 0.937, and decay of 0.0005. Given the memory of the graphics card we use, we set the batch_size to 16. Here, a stochastic gradient descent (SGD) was used as the optimizer with a cosine learning rate decay strategy. All other parameters were set to the YOLOv5s’ default values.

### 5.3. Evaluation Metrics

We evaluated different models in terms of precision, recall, F1-score, and mean average precision (mAP). Precision (P) is defined as the ratio of true positive detections to the total number of positive detections made by the model. A higher precision indicates a lower likelihood of false positive detections; thus, it is also referred to as accuracy. In contrast, recall (R) is the ratio of true positive detections to the total number of actual positive samples. Precision and recall are calculated as follows:(13)$$Precision=\frac{TP}{TP+FP}$$
(14)$$Recall=\frac{TP}{TP+FN}$$
here, TP (true positive) represents the samples that are positive and predicted correctly as positive by the model, TN (true negative) represents the samples that are negative and predicted correctly as negative by the model, and FP (false positive) represents the samples that are negative but predicted incorrectly as positive by the model. Finally, FN (false negative) represents the samples that are positive but predicted incorrectly as negative by the model.

Equations (13) and (14) can be used to calculate the precision and recall values at various thresholds, which are then used to plot the P-R curve. The area under the P-R curve and the coordinate axes represents the average precision (AP), which is calculated as follows:(15)$$AP=\int_{0}^{1}P\left(R\right)dR$$

The area under the P-R curve represents the AP for each class, and the mAP is the average of the AP values across all classes. The mAP is calculated as follows:(16)$$mAP=\frac{\sum_{i=1}^{k}AP_{i}}{k}$$

The F1-score is the harmonic mean of the precision and recall, where an equal weight is given to both metrics. The F1-score is calculated as follows:(17)$$F_{1}=\frac{2\cdot Precision\cdot Recall}{Precision+Recall}$$

To assess the extent to which the model is lightweight, the experiments also considered the network model’s parameter count, the size of the model’s weight, and the number of floating-point operations (FLOPs). Note that these metrics are inversely related to the model’s lightweight nature, and FPS (frames per second) describes the algorithm’s inference speed.

### 5.4. Experimental Results

The training process of the proposed method is shown in Figure 10. As can be seen, over the same 100 training epochs, the loss of the proposed method decreases more quickly.

Table 3 compares the performance of the proposed method and the YOLOv5s model. As shown, the proposed method improves mAP@0.5 by 3.9% and mAP@0.5:0.95 by 3.3%. In addition, the proposed method reduces the number of parameters by 71%, and it reduces FLOPs by 58% compared to YOLOv5s.

The FPS of our method is 20 lower than that of YOLOv5s. As shown in Table 4, the main increase in time occurs during the prediction frame generation stage due to the use of large convolution kernels (5 × 5 and 7 × 7) in the split-DLKA module, which lead to longer inference times. We traded off some inference speed to achieve higher detection accuracy.

Figure 11 and Figure 12 show the P-R curves of the two models. As can be seen, the proposed method achieves a better balance between precision and recall, as well as a higher mAP value.

Figure 13 and Figure 14 show the confusion matrices of the two models. Here, it is evident that the proposed method achieves higher accuracy than YOLOv5s when detecting ore carriers, general cargo ships, bulk cargo carriers, and container ships.

The results obtained on the test set, as shown in Figure 15, highlight the proposed model’s superior detection capabilities for various ship types. For example, in the fourth row of images, YOLOv5s exhibits false positives for the same ship and misses the detection of a small boat. In contrast, the proposed method demonstrates no false positives and performs well when detecting small vessels.

In the Grad-CAM++ heat map (Figure 16), darker colors represent areas that contribute more significantly to ship detection. In the heat map generated by YOLOv5s, the color distribution is relatively dispersed, with many high-intensity regions appearing in the background. This suggests that, in addition to focusing on the target object, YOLOv5s also allocates unnecessary attention to the background during detection. In contrast, our approach demonstrates clearer target edges (such as the ship’s outline) in the heat maps, with colors more concentrated around the target’s shape, indicating higher confidence in the detection of the target object.

### 5.5. Comparison with Other Algorithms

We also selected several single-stage object detection networks for comparison, and the corresponding results are shown in Table 5.

Table 5 compares several popular single-stage object detection algorithms based on their performance metrics. As can be seen, for ship detection tasks, the YOLOv9c, YOLOv5m, and YOLOv5l models exhibit superior performance. However, these networks exceed the parameters and computational costs of the proposed method considerably. For example, YOLOv9c has 12 times more parameters, 15.9 times more FLOPs, and 11.7 times more weight size to the proposed method. Despite these differences, the proposed method achieves competitive performance, trailing YOLOv9c by 7% in terms of mAP@0.5. Generally, a larger weight size implies that more memory and computational power are required to run the network, which is a significant drawback for unmanned surface vessels. Among the single-stage object detection networks of similar parameter levels, the proposed method has fewer parameters and FLOPs than YOLOv4-tiny while achieving the highest mAP. Compared with YOLOv9c, the proposed method strikes an effective balance between detection accuracy and model size, with a 7% decrease in mAP@0.5, a 91% reduction in the number of parameters, a 93% reduction in FLOPs, and a 12 ms faster inference time.

### 5.6. Ablation Experiment

First, we validated the improvement in the network realized by changing the activation function in ShuffleNetV2 from ReLU to LeakyReLU.

Table 6 shows that replacing the original backbone network with ShuffleNetV2 results in a 74% reduction in the number of parameters and a 75% reduction in FLOPs. However, this change leads to a slight decline in the precision, recall, and mAP values. To enhance detection performance, the activation function of the backbone network was switched to LeakyReLU, which resulted in a 1.2% increase in mAP@0.5 compared to ReLU.

An ablation study was conducted using YOLOv5s as the baseline model to validate the effectiveness of the proposed method. Here, the models were compared in terms of parameters, FLOPs, and mAP. For convenience, the version of ShuffleNetV2 with LeakyReLU replacing ReLU is referred to as LeakyShuffle. The corresponding experimental results are shown in Table 7.

The following conclusions can be drawn from the results in Table 7: 

The proposed method achieves the best performance in terms of mAP. Compared to the baseline YOLOv5s model, the proposed method increases mAP@0.5 by 3.9% and mAP@0.5:0.95 by 3.3%. Each of the three proposed improvements contribute to the increase in the mAP value. Replacing the backbone with ShuffleNetV2 primarily reduces the model’s parameter count and computational load by 74% and 75%, respectively; meanwhile, it also results in a slight mAP increase. The results of the third ablation experiment demonstrate the effectiveness of the proposed split-DLKA attention module. Ultimately, the final method also achieves the highest F1-score.

To further demonstrate the effectiveness of the split-DLKA in terms of parameter count reduction and detection accuracy improvement, we conducted comparative experiments with other attention mechanisms using YOLOv5s as the baseline. The results are shown in Table 8.

As can be seen, among the various attention mechanisms, split-DLKA shows the greatest improvement in terms of both mAP@0.5 and mAP@0.5:0.95. The increased computational effort of split-DLKA leads to a reduction in FPS; however, it achieves higher detection accuracy. Some of the FPS loss can be mitigated by optimizing the backbone network. In essence, split-DLKA offers a better trade-off between detection accuracy and speed. To determine the most suitable position for adding the attention layer, we conducted three comparative experiments using YOLOv5s with the replaced backbone network as the baseline (Figure 17). The results demonstrate that adding the split-DLKA at position 1 performs best in terms of detection performance, as shown in Table 9.

The impact of hyperparameters on experimental results is shown in Table 10. As can be seen from the table, when α=1.9,δ=3, the model has the highest detection accuracy.

Adding the split-DLKA at position 1 yields the best results because the layers following position 1 are utilized by the detection head to detect small objects. In this case, split-DLKA processes a feature map with fewer channels compared to the other two positions, resulting in a higher FPS. Typically, small objects are the most challenging to detect in object detection tasks. This indicates that the proposed split-DLKA can improve the accuracy of small object detection significantly.

## 6. Conclusions

Ship object detection requires a detection network with a low parameter count and computational load to meet hardware requirements. To reduce the computational costs of the object detection network while improving detection accuracy, this paper proposed an improved lightweight single-stage object detection network. By replacing the original backbone network with ShuffleNetV2, the network parameter count and computational load of the proposed method are reduced considerably. In addition, various enhancements, e.g., improvements to the activation and loss functions, are implemented in the proposed method to improve detection accuracy.

This paper also designed a new large-kernel deformable convolution attention mechanism. The network’s performance is enhanced by leveraging the adaptability of deformable convolution to samples of different shapes and the larger receptive field of the large-kernel convolution. Experimental results demonstrate that compared to the baseline model, the proposed method achieves improvements in mAP@0.5, mAP@0.5:0.95, precision, and recall by 3.9%, 3.3%, 5.9%, and 1.2%, respectively. In addition, the proposed method demonstrates superior performance in terms of both detection accuracy and model size compared to other single-stage object detection algorithms.

However, our proposed method has a slower inference time compared to YOLOv5s and has not been tested under complex meteorological conditions. Future work should include real ship validation experiments and evaluate the detection model’s performance in challenging weather scenarios. With advancements in unmanned surface vessel hardware technology, multiple cameras will be installed on ships, generating large-scale multi-view data. In such scenarios, it is crucial to significantly reduce data size [39] and extract features from multi-view data [40]. This is also a key direction for future research.

## Figures and Tables

**Figure 1 sensors-24-05603-f001:**
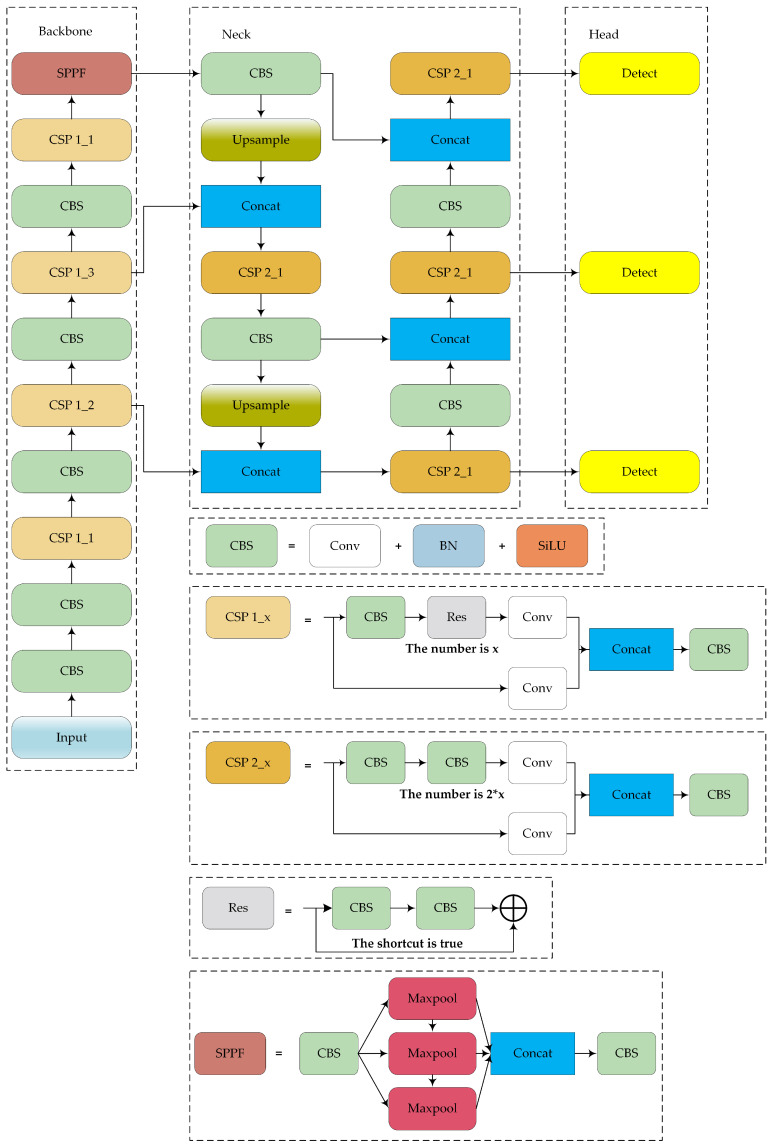
Structure of YOLOv5s.

**Figure 2 sensors-24-05603-f002:**
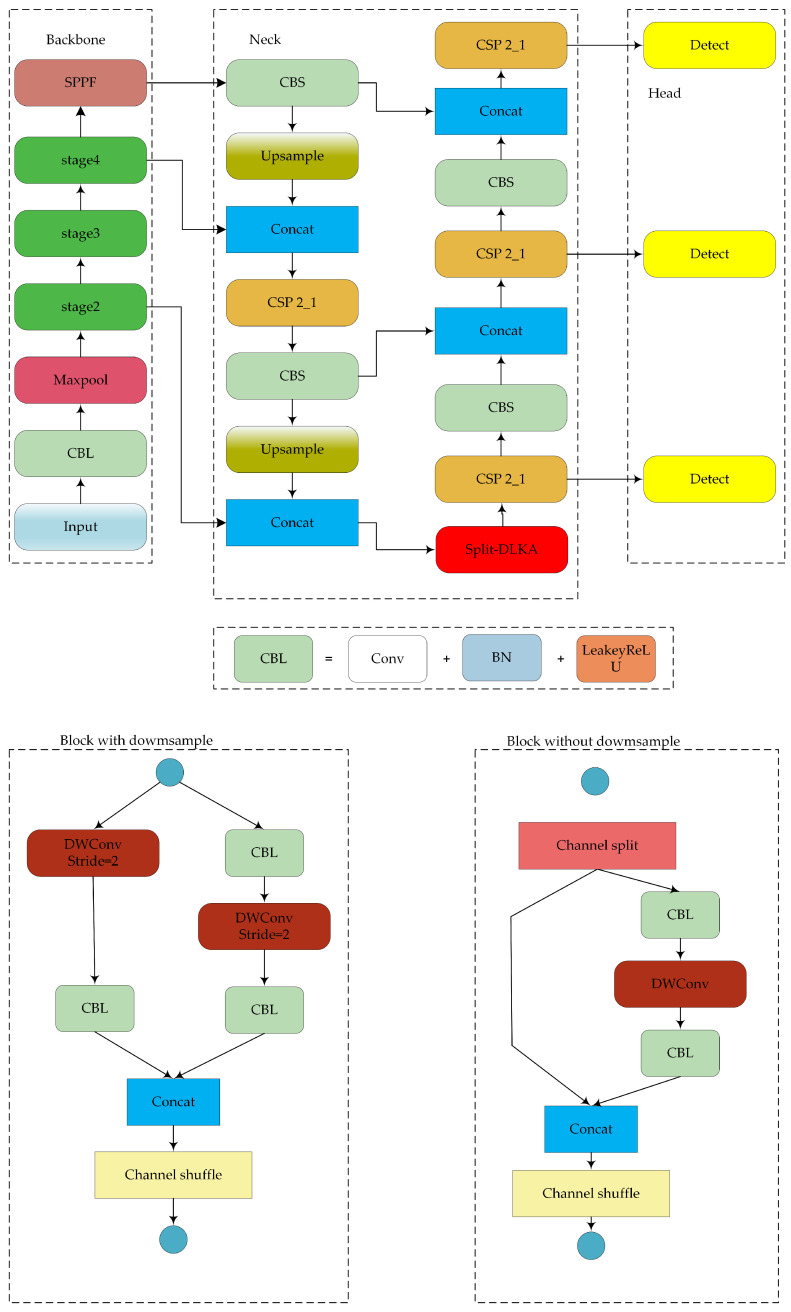
Structure of the proposed model.

**Figure 3 sensors-24-05603-f003:**
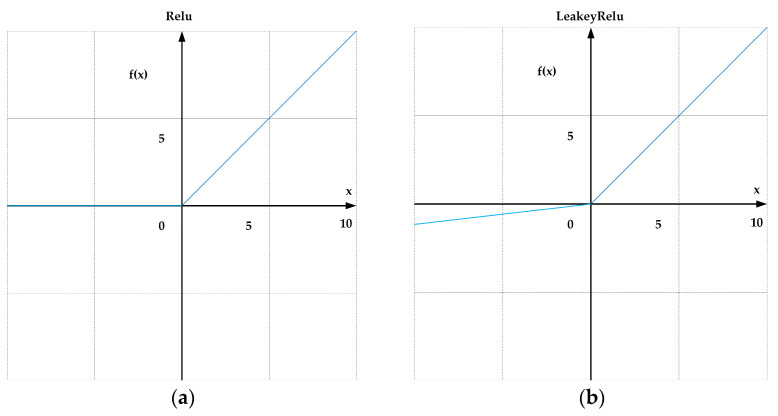
(**a**) ReLU activation function plot. (**b**) LeakyReLU activation function plot.

**Figure 4 sensors-24-05603-f004:**
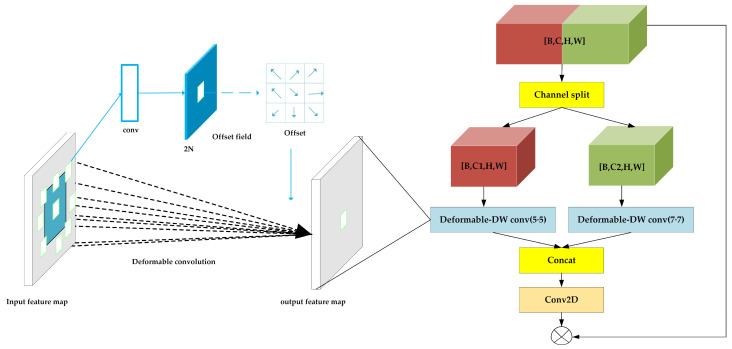
Structure of split-DLKA.

**Figure 5 sensors-24-05603-f005:**
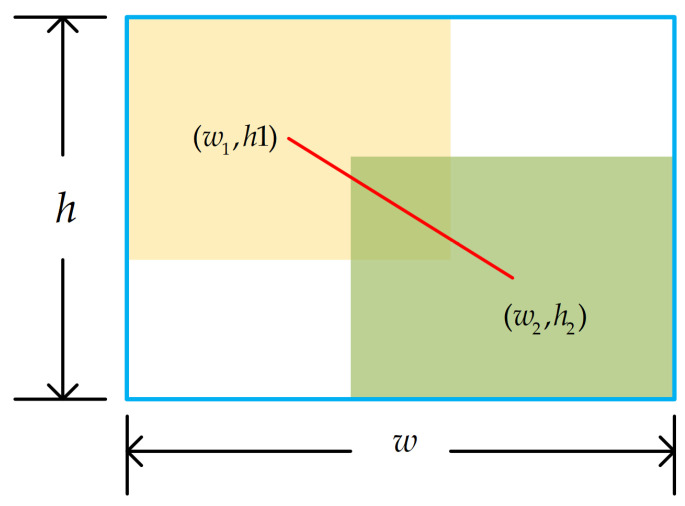
Ground truth, predicted box, and their minimum enclosing rectangle.

**Figure 6 sensors-24-05603-f006:**
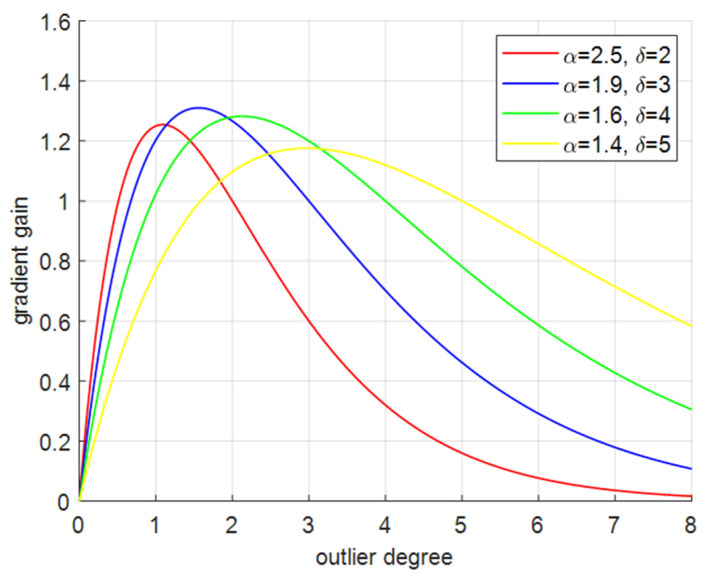
Curves of outlier degree and gradient gain under different hyperparameters.

**Figure 7 sensors-24-05603-f007:**
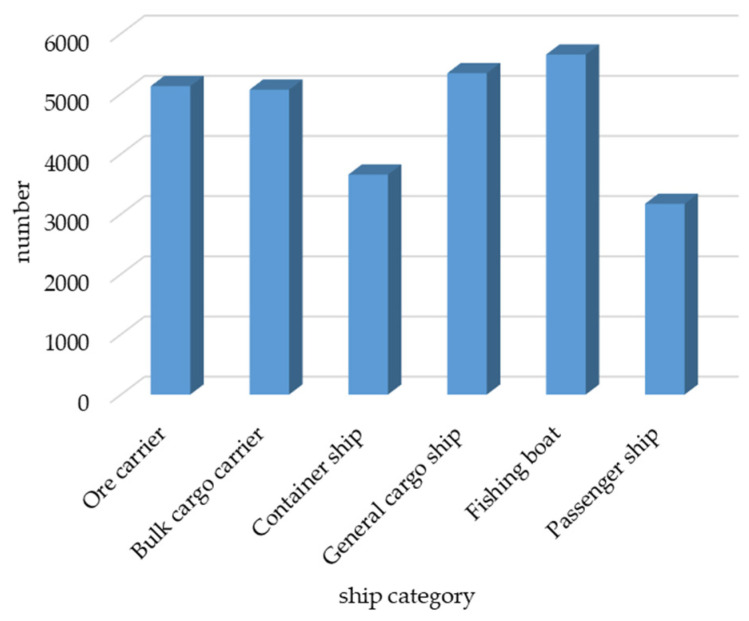
Types of ships and their quantities.

**Figure 8 sensors-24-05603-f008:**
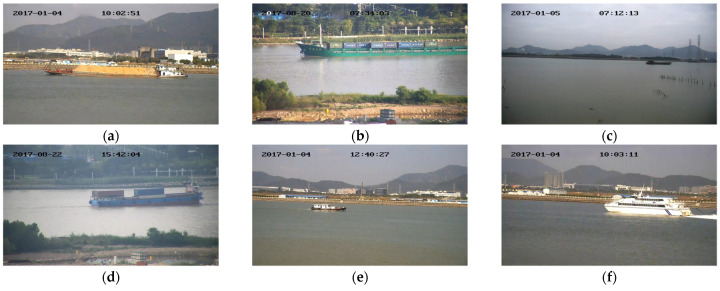
(**a**) Ore carrier; (**b**) general cargo ship; (**c**) bulk cargo carrier; (**d**) container ship; (**e**) fishing boat; and (**f**) passenger ship.

**Figure 9 sensors-24-05603-f009:**
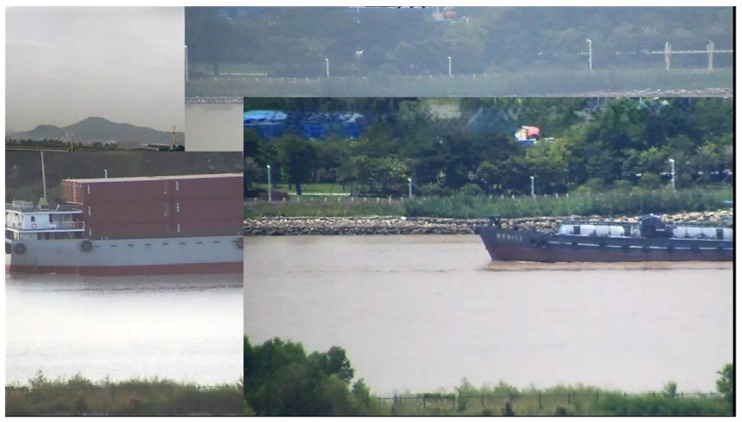
Augmented images.

**Figure 10 sensors-24-05603-f010:**
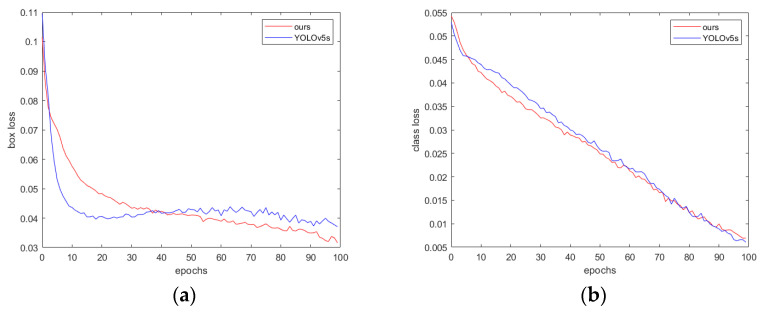
(**a**) Box loss: measuring the difference between the predicted box position and the true label position; (**b**) class loss: measuring the accuracy of classifying different object categories; (**c**) object loss: measuring the accuracy of determining the presence of an object in the image; (**d**) F1 score; (**e**) mAP@0.5; (**f**) mAP@0.5:0.95; (**g**) precision; and (**h**) recall.

**Figure 11 sensors-24-05603-f011:**
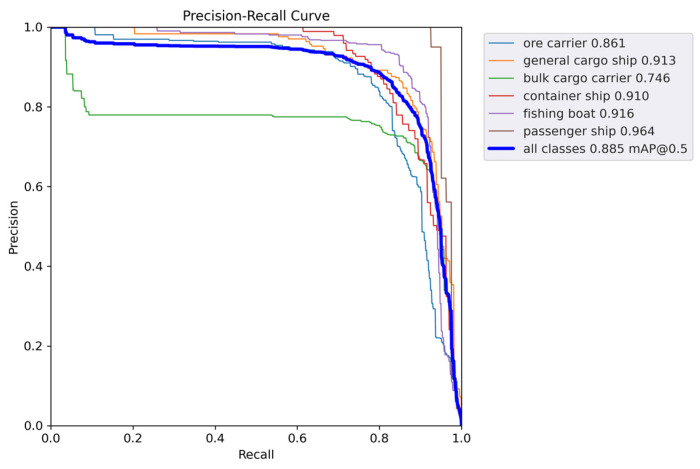
mAP@0.5 curve of YOLOv5s.

**Figure 12 sensors-24-05603-f012:**
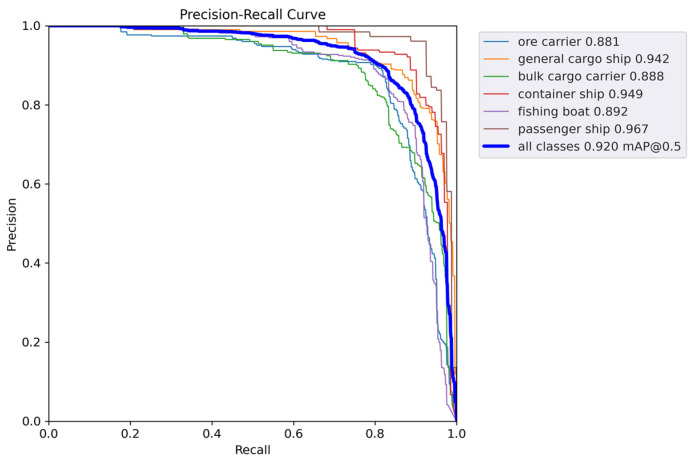
mAP@0.5 curve of the proposed method.

**Figure 13 sensors-24-05603-f013:**
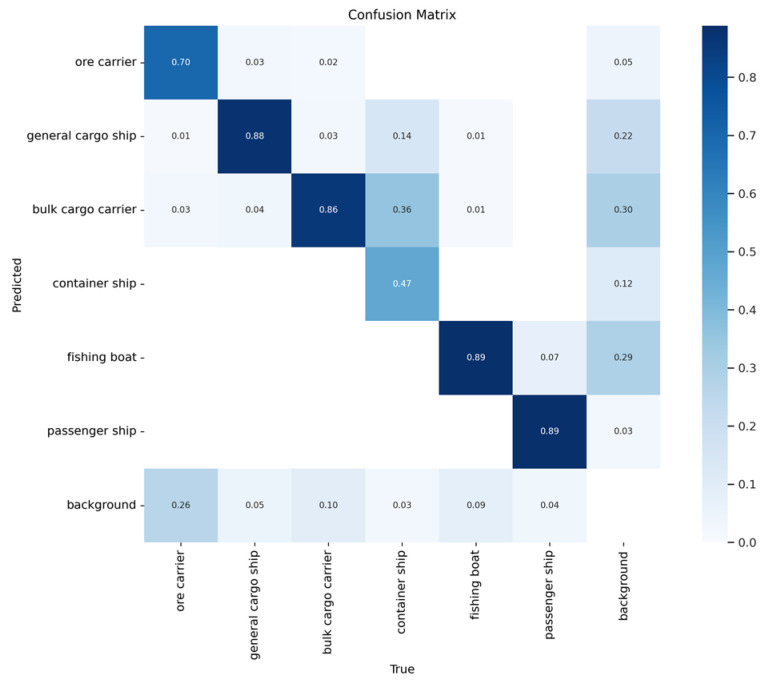
Confusion matrix of YOLOv5s.

**Figure 14 sensors-24-05603-f014:**
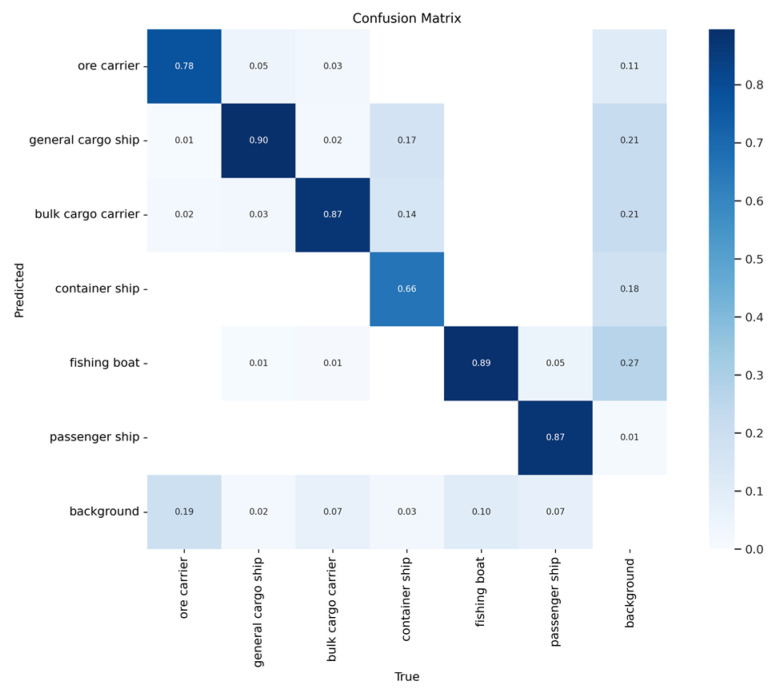
Confusion matrix of the proposed method.

**Figure 15 sensors-24-05603-f015:**
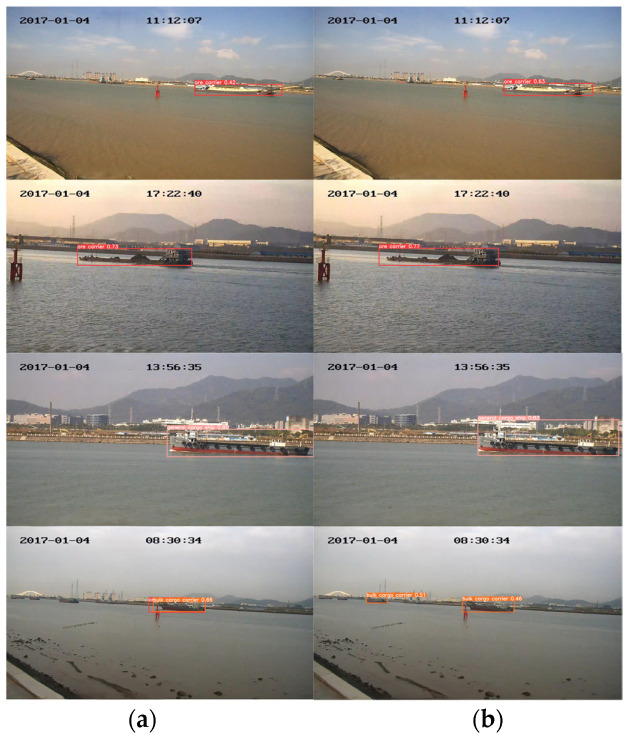
(**a**) YOLOv5s detection results; (**b**) the proposed method’s detection results.

**Figure 16 sensors-24-05603-f016:**
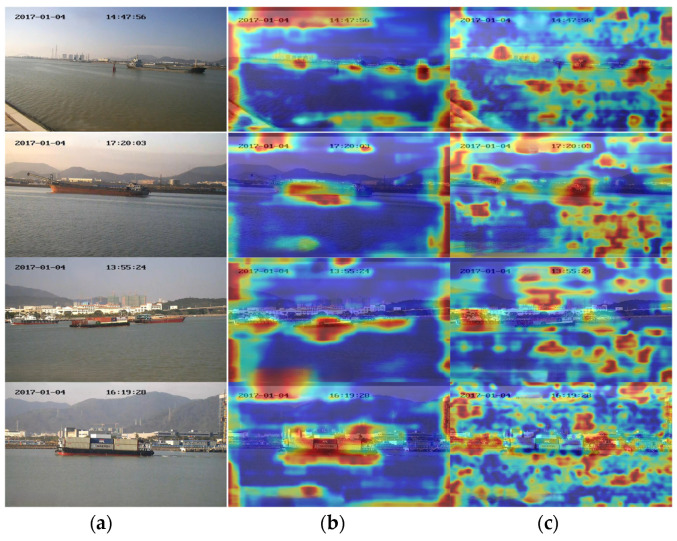
(**a**) Original image; (**b**) heatmap of the proposed method; and (**c**) heatmap of YOLOv5s.

**Figure 17 sensors-24-05603-f017:**
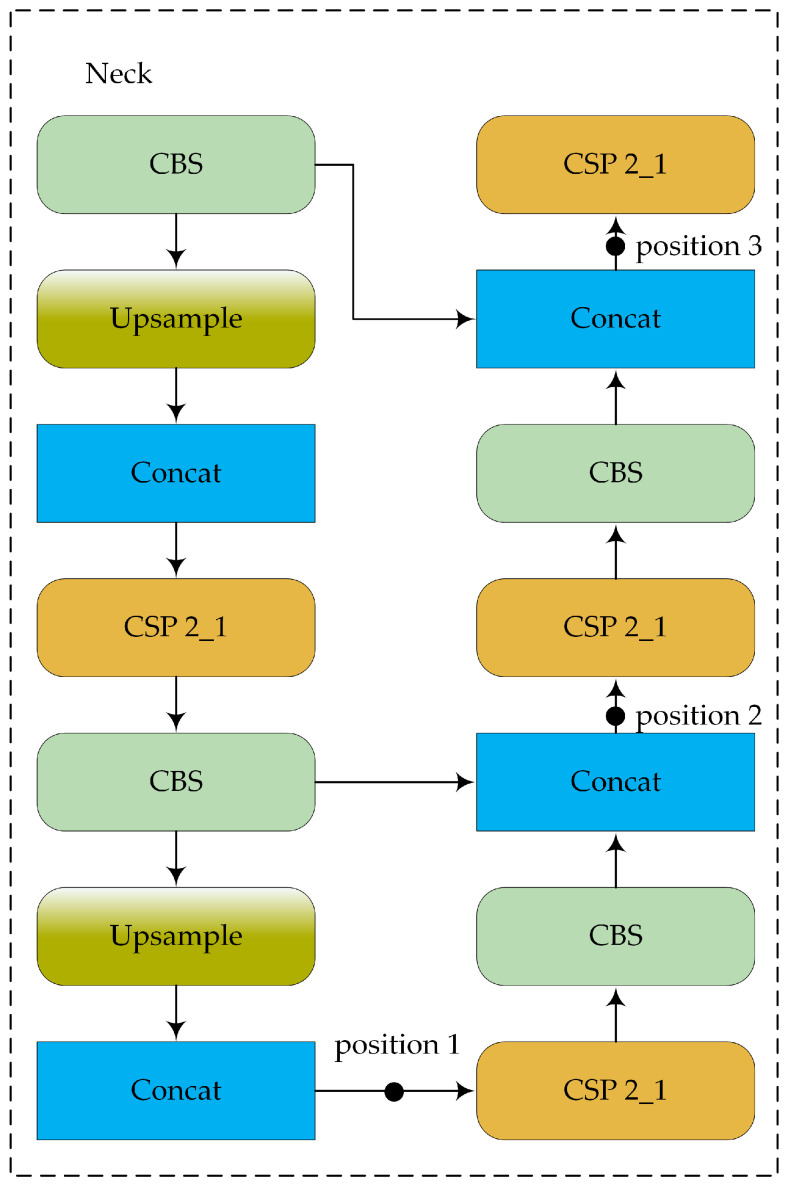
Illustration of different insertion positions.

**Table 1 sensors-24-05603-t001:** ShuffleNetV2 structure parameters.

Layer	Output Size	Kernel Size	Stride	Repeat	Channels
Input	224 × 224				3
Conv1	112 × 112	3 × 3	2	1	24
MaxPool	56 × 56	3 × 3	2	1	24
Stage 2	28 × 28		2 1	1 3	116
Stage 3	14 × 14		2 1	1 3	232
Stage 4	7 × 7		2 1	1 3	464

**Table 2 sensors-24-05603-t002:** Experimental platform and environment configuration.

Projects	Environment
CPU	12th Gen Intel(R) Core (TM) i7-12700F
GPU	NVIDIA GeForce RTX 3060
RAM	16 G
CUDA	12.1
Framework	Pytorch 2.1.2

**Table 3 sensors-24-05603-t003:** Comparison of performance between YOLOv5 and the proposed method.

Evaluation Metrics	YOLOv5s	Ours
Parameters	7.04 M	2.02 M
FLOPs	15.8 G	6.5 G
Weight size	14.4 MB	4.4 MB
Precision	0.84	0.89
Recall	0.83	0.84
F1	0.841	0.864
mAP@0.5	0.885	0.920
mAP@0.5:0.95	0.545	0.563
FPS (frame/s)	212	192

**Table 4 sensors-24-05603-t004:** Analysis of inference time between YOLOv5s and our method.

Method	Preprocess	Inference	NMS
YOLOv5s	0.3 ms	3.7 ms	0.7 ms
ours	0.3 ms	4.1 ms	0.8 ms

Preprocess means affine transformation; inference means generating prediction boxes; and NMS is used to remove redundant prediction boxes.

**Table 5 sensors-24-05603-t005:** Contrast experiments.

Model	Parameters	FLOPs	Precision	Weight Size	Recall	F1	mAP@0.5	mAP@0.5:0.95	FPS
YOLOv5m	20.89 M	48.3 G	0.879	42.2 MB	0.88	0.879	0.944	0.602	119
YOLOv5l	46.17 M	108.3 G	0.889	92.8 MB	0.89	0.889	0.946	0.609	66
yolov4-tiny	5.89 M	6.8 G	0.878	24.3 MB	0.45	0.595	0.689	0.303	370
Nanodet-plus	0.95 M	1.2 G	0.667	2.44 MB	0.62	0.642	0.641	0.341	55
YOLOv5s-efficientnetv2	5.6 M	5.6 G	0.738	11.5 MB	0.74	0.738	0.809	0.431	208
YOLOv9c	25 M	103.7 G	0.98	51.8 MB	0.97	0.974	0.99	0.82	58
ours	2.02 M	6.5 G	0.89	4.4 MB	0.84	0.864	0.920	0.563	192

**Table 6 sensors-24-05603-t006:** Contrast experiment between ReLU and LeakyReLU activation functions.

Model	Parameters	FLOPs	Precision	Recall	mAP@0.5	mAP@0.5:0.95
YOLOv5s	7.04 M	15.8 G	0.84	0.83	0.885	0.545
YOLOv5s + shufflenetv2	1.8 M	3.9 G	0.79	0.82	0.877	0.504
YOLOv5s + shufflenetv2 (LeakeyReLu)	1.8 M	3.9 G	0.85	0.79	0.888	0.508

**Table 7 sensors-24-05603-t007:** Ablation experiment.

LeakeyShuffle	Split-DLKA	WIOU	Parameters	FLOPs	Precision	Recall	mAP@0.5	mAP@0.5:0.95	F1
			7.04 M	15.8 G	0.84	0.83	0.885	0.545	0.834
√			1.8 M	3.9 G	0.85	0.79	0.888	0.508	0.818
	√		7.4 M	21.2 G	0.84	0.84	0.907	0.552	0.840
		√	7.04 M	15.8 G	0.85	0.84	0.909	0.549	0.844
√	√		2.02 M	6.5 G	0.85	0.84	0.905	0.537	0.844
√		√	1.8 M	3.9 G	0.82	0.79	0.855	0.480	0.804
	√	√	7.4 M	21.2 G	0.86	0.82	0.919	0.529	0.839
√	√	√	2.02 M	6.5 G	0.89	0.84	0.920	0.563	0.864

√ means the improvement was used.

**Table 8 sensors-24-05603-t008:** Comparison of different attention mechanisms.

Attention Mechanism	Parameters	FLOPs	FPS	Precision	Recall	mAP@0.5	mAP@0.5:0.95
SE	7.2 M	16.6 G	204	0.83	0.84	0.900	0.545
CA	7.2 M	16.7 G	204	0.82	0.89	0.896	0.519
ECA	7.2 M	16.6 G	217	0.82	0.85	0.901	0.538
simAM	7.2 M	16.6 G	204	0.84	0.83	0.896	0.529
Split-DLKA	7.4 M	21.2 G	188	0.84	0.84	0.907	0.552

**Table 9 sensors-24-05603-t009:** Experiments with split-DLKA at different insertion positions.

Position	Parameters	FLOPs	FPS	Precision	Recall	mAP@0.5	mAP@0.5:0.95
Position 1	2.02 M	6.5 G	192	0.85	0.84	0.905	0.537
Position 2	2.2 M	5.2 G	121	0.80	0.79	0.870	0.497
Position 3	2.2 M	5.2 G	69	0.77	0.82	0.867	0.498

**Table 10 sensors-24-05603-t010:** Effects of different hyperparameters on model performance.

α	δ	Precision	Recall	mAP@0.5	mAP@0.5:0.95
2.5	2	0.84	0.82	0.891	0.539
1.9	3	0.85	0.85	0.912	0.560
1.6	4	0.83	0.85	0.901	0.555
1.4	5	0.83	0.84	0.894	0.545

## Data Availability

The data presented in this study are available in SeaShips at https://github.com/jiaming-wang/SeaShips.git (accessed on 24 March 2024).

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
