# Peer review of "Lightweight Single-Stage Ship Object Detection Algorithm for Unmanned Surface Vessels Based on Improved YOLOv5"

_sensors, 2024, doi:10.3390/s24175603_

Round 1
Reviewer 1 Report
Comments and Suggestions for Authors
In this paper, an attention-mechanism-based target detection algorithm for light single-stage surface ships is proposed. To reduce the number of model parameters and the amount of computation, we used an enhanced ShuffleNetV2 network as the backbone. In addition, a separate DLKA module is designed and implemented in the small target detection layer to improve the detection accuracy. Finally, the WIOUv3 loss function is introduced to minimize the influence of low-quality samples on the model. However, there are some questions that need to be revised or answered:
1. The method proposed in this paper is based on the yolov5 network, including model interpretation and experimental comparison, which focus on the comparison with yolov5. It is suggested to modify the title of the paper and make improvements on yolov5, otherwise it is necessary to supplement the experimental part and description.
2. Section4.1 needs to explain why we replaced ReLU with LeakyReLU activation function.
3. split-DLKA proposed in Section4.2.2, which aims to improve the multi-scale sensing capability of the network, is not directly related to the screening of anchor frame size, and it is recommended to consider the description in this paragraph.
4. batch_size is a very important parameter in deep learning. In the part of experimental parameter setting, it is best to explain the size of batch_size.
5. The visualization effect of heat map in Figure 15 is not obvious, so it is recommended to try other visualization methods of heat map.
6. Some related works should be discussed, including, "Multiple information perception-based attention in YOLO for underwater object detection", "Manifold-based Incomplete Multi-view Clustering via Bi-Consistency Guidance" and "Graph-Collaborated Auto-Encoder Hashing for Multi-view Binary Clustering".
7. The comparison results of frame numbers are added in Table 7 and Table 8 to confirm the superior lightweight effect of the method proposed in this paper.
Comments on the Quality of English Language
English writting is good.
Author Response
Comments 1: The method proposed in this paper is based on the yolov5 network, including model interpretation and experimental comparison, which focus on the comparison with yolov5. It is suggested to modify the title of the paper and make improvements on yolov5, otherwise it is necessary to supplement the experimental part and description.
Response 1: We are grateful for your inquiry. As you suggested, we have revised the title of our article to: “Lightweight Single-Stage Ship Object Detection Algorithm for Unmanned Surface Vessels Based on Improved YOLOv5.” We believe this new title more accurately reflects the focus of our work and addresses your feedback effectively.
Comments 2: Section4.1 needs to explain why we replaced ReLU with LeakyReLU activation function.
Response 2: Thank you for raising this question. We replaced the ReLU activation function with the Leaky ReLU function in our proposed algorithm for several reasons:
ReLU can suffer from the "dying ReLU" problem, where neurons become inactive and stop learning if their output is zero for all inputs. Leaky ReLU has been shown to improve the performance of deep networks by maintaining a more stable gradient flow. This can lead to faster convergence and better overall accuracy in object detection tasks.
To prevent reader confusion, based on your feedback, we have included the following content in the paper:
Page 8 at line 258 to 269: ReLU is a single-sided saturation function with robustness against noise interference. ReLU truncates negative values to zero, thereby introducing sparsity and improving computational efficiency (Figure 3 a). However, ReLU activation function outputs zero for negative input values, which can lead to neurons becoming "dead" or inactive during training, meaning their gradients are consistently zero and they cannot contribute to learning. The Leaky ReLU function addresses this issue by allowing a small, non-zero slope in the negative region, ensuring that all neurons remain partially active and continue to update during training. By preserving a small gradient for negative inputs, Leaky ReLU helps maintain gradient flow throughout the network, which enhances both the stability and convergence speed of the training process. This enhancement is particularly crucial for our object detection tasks, as these models often require deep network architectures to effectively capture complex features. The formula for LeakyReLU is given as follows:
Page 8 at line 273 to 279: When the input to the activation function is less than zero, the gradient can still be calculated. In our experiments, we observed that Leaky ReLU outperforms ReLU in terms of performance. Specifically, the use of Leaky ReLU resulted in improved detection accuracy of the model. Experimental results have shown in Table 6 that in the ship object detection task, replacing the activation function of ShuffleNetV2 with LeakyReLU and using it as the backbone of YOLOv5s can yield a 0.3% increase in mAP@0.5 and a 7.7% increase in precision.
Comments 3: split-DLKA proposed in Section4.2.2, which aims to improve the multi-scale sensing capability of the network, is not directly related to the screening of anchor frame size, and it is recommended to consider the description in this paragraph.
Response 3: Thank you for pointing out this error. Your suggestion is very reasonable, and with your suggestion in mind, we removed the description of anchor and rewrote the section as follows:
Page 8 at line 279 to 287: In the task of ship target detection, the size and shape of different ship types can vary significantly, encompassing a wide range of morphologies and structures. This diversity necessitates that detectors be capable of adapting to multiple forms, which increases model complexity and makes it challenging to generalize across different ship types. Additionally, the maritime environment, including oceans, harbors, and other water contexts, is often complex and dynamic, with potential disturbances such as waves, other vessels, and infrastructure. Moreover, the diversity in ship sizes and shapes can lead to sample imbalance in the training data, resulting in suboptimal model performance for certain ship categories.
Comments 4: batch_size is a very important parameter in deep learning. In the part of experimental parameter setting, it is best to explain the size of batch_size.
Response 4: Apologies for this question. We have added the batch_size parameter:
Page 14 at line 396 to 397:In this experiment, training was performed over 100 epochs with a learning rate of 0.01, momentum of 0.937, and decay of 0.0005. Given the memory of the graphics card we use, setting batch_size to 16.
Comments 5: The visualization effect of heat map in Figure 15 is not obvious, so it is recommended to try other visualization methods of heat map.
Response 5: Thank you for your query. As suggested, Grad-CAM may not be the most suitable method for comparing the performance of the two models. Therefore, I have replaced Grad-CAM with Grad-CAM++, which incorporates second-order gradients and third-degree steps for a more refined analysis. As a result, both the image and its corresponding interpretation have been updated accordingly.
Page 15 at line 450 to 457:In the Grad-CAM heat map (Figure 16), darker colors represent areas that contribute more significantly to ship detection. In the heat map generated by YOLOv5s, the color distribution is relatively dispersed, with many high-intensity regions appearing in the background. This suggests that, in addition to focusing on the target object, YOLOv5s also allocates unnecessary attention to the background during detection. In contrast, our approach demonstrates clearer target edges (such as the ship's outline) in the heat maps, with colors more concentrated around the target's shape, indicating higher confidence in the detection of the target object.
Page 21 Figure 16:

Comments 6: Some related works should be discussed, including, "Multiple information perception-based attention in YOLO for underwater object detection", "Manifold-based Incomplete Multi-view Clustering via Bi-Consistency Guidance" and "Graph-Collaborated Auto-Encoder Hashing for Multi-view Binary Clustering".
Response 6: Thank you for your suggestion. The work you mentioned is highly relevant to our research. I have cited the first work in the related work section, while the second and third works are cited in the conclusion section, where they are proposed as potential directions for future improvements.
Page 3 at line 105 to 111: For instance, Shen et al [14]. introduced a Multiple Information Perception-based Attention Module (MIPAM). Their approach incorporates channel-level information col-lection through global covariance pooling and channel-wise global average pooling, while spatial-level information is gathered via spatial-wise global average pooling and cross-spatial global covariance pooling. This method enriches feature representation, leading to improved detection accuracy when integrated into the YOLO detector.
Page 22 at line 559 to 562: With advancements in unmanned surface vessels hardware technology, multiple cameras will be installed on ships, generating large-scale multi-view data. In such scenarios, it is crucial to significantly reduce data size [39] and extract features from multi-view data [40]. This is also a key direction for future research.
Comments 7: The comparison results of frame numbers are added in Table 7 and Table 8 to confirm the superior lightweight effect of the method proposed in this paper.
Response 7: Thank you for bringing up this question. As per your suggestion, we have included the FPS metric in the table and have discussed and analyzed the results accordingly. With the addition of the new table, the former Tables 7 and 8 are now renumbered as Tables 8 and 9, respectively. The updated content is as follows:
Page 21 at line 521 to 530:
Table 8. Comparison of different attention mechanisms.
|
Attention mechanism |
Parameters |
FLOPs |
FPS |
Precision |
Recall |
mAP@0.5 |
mAP@0.5:0.95 |
|
SE |
7.2M |
16.6G |
204 |
0.83 |
0.84 |
0.900 |
0.545 |
|
CA |
7.2M |
16.7G |
204 |
0.82 |
0.89 |
0.896 |
0.519 |
|
ECA |
7.2M |
16.6G |
217 |
0.82 |
0.85 |
0.901 |
0.538 |
|
simAM |
7.2M |
16.6G |
204 |
0.84 |
0.83 |
0.896 |
0.529 |
|
Split-DLKA |
7.4M |
21.2G |
188 |
0.84 |
0.84 |
0.907 |
0.552 |
As can be seen, among the various attention mechanisms, split-DLKA shows the greatest improvement in terms of both mAP@0.5 and mAP@0.5:0.95. The increased computational effort of split-DLKA leads to a reduction in FPS; however, it achieves higher detection accuracy. Some of the FPS loss can be mitigated by optimizing the backbone network. In essence, split-DLKA offers a better trade-off between detection accuracy and speed.
Page 22 at line 531 to 536:
Table 9. Experiments with split-DLKA at different insertion positions.
|
Position |
Parameters |
FLOPs |
FPS |
Precision |
Recall |
mAP@0.5 |
mAP@0.5:0.95 |
|
Position 1 |
2.02M |
6.5G |
192 |
0.85 |
0.84 |
0.905 |
0.537 |
|
Position 2 |
2.2M |
5.2G |
121 |
0.80 |
0.79 |
0.870 |
0.497 |
|
Position 3 |
2.2M |
5.2G |
69 |
0.77 |
0.82 |
0.867 |
0.498 |
Adding the split-DLKA at position 1 yields the best results because the layers following position 1 are utilized by the detection head to detect small objects. In this case, split-DLKA processes a feature map with fewer channels compared to the other two positions, resulting in higher FPS. Typically, small objects are the most challenging to detect in object detection tasks. This indicates that the proposed split-DLKA can improve the accuracy of small object detection significantly.
Response to Comments on the Quality of English Language
Point 1: Minor editing of English language required.
Response 1: Thanks for your suggestion, we have revised the manuscript according to the polishing comments of professional institutions. The following is the proof of polishing.

Reviewer 2 Report
Comments and Suggestions for Authors
This paper introduces a lightweight, single-stage ship target detection algorithm for surface unmanned boats that incorporates an attention mechanism, offering both theoretical significance and practical value. The algorithm employs an enhanced ShuffleNetV2 network as its backbone. Additionally, a novel separated DLKA module has been designed and implemented in the small target detection layer to enhance detection precision. The introduction of the WIOUv3 loss function aims to minimize the impact of low-quality samples on the model. The algorithm maintains detection accuracy while reducing the number of parameters and computational requirements, demonstrating a degree of innovation. However, the paper has several issues that need to be addressed:
1)The introduction section does not adequately explain the rationale for choosing YOLOv5s as the baseline model, nor does it clarify why more recent versions like YOLOv8n or YOLOv10n were not considered.
2)The second chapter, which reviews related work, merely lists previous studies without a coherent logical structure that would aid the reader's comprehension.
3)The attention mechanism section in the fourth chapter lacks emphasis on key points. The title of section 4.2 should be "Attention DLKA Module," with a detailed explanation of this module, while the discussion on deformable convolution could be reduced.
4)The section on the loss function in the fourth chapter should include diagrams to improve readability and help readers understand the concepts better.
5)The selection of hyperparameters for the loss function in the fourth chapter is missing comparative analysis experiments.
6)In Table 3, which compares the algorithm's performance with YOLOv5, the algorithm has fewer parameters and less computational load than YOLOv5, yet it has a lower FPS, and this discrepancy should be explained.
7)The evaluation metrics should include the final model's weight size to highlight the ease of deployment on edge devices.
Comments on the Quality of English Language
1)The grammar and spelling of the manuscript need further improvement.
Author Response
Comments 1: The introduction section does not adequately explain the rationale for choosing YOLOv5s as the baseline model, nor does it clarify why more recent versions like YOLOv8n or YOLOv10n were not considered.
Response 1: We are grateful for your inquiry. The reason we chose YOLOv5 is that it has proven its reliability across a variety of scenarios after extensive practice and optimization. Additionally, YOLOv5 benefits from strong community support, comprehensive documentation, and numerous tutorials, making debugging and optimization more convenient. At the time this work began, Ultralytics had not yet released an official version of YOLOv10, so it was not considered as a baseline model. In the introduction section, we have explained in detail why YOLOv5 was selected. The reasoning is as follows:
Page 2 at line 75 to 80: The reasoning is that, after extensive practice and optimization, YOLOv5 has proven its reliability across a wide range of scenarios. Thanks to its prolonged development period, the YOLOv5 model is relatively stable, with fewer potential issues. As the smaller variant in the YOLOv5 series, YOLOv5s strikes an excellent balance between performance and efficiency. It performs well in resource-constrained environments, maintaining high detection accuracy while ensuring faster inference speeds.
Comments 2: The second chapter, which reviews related work, merely lists previous studies without a coherent logical structure that would aid the reader's comprehension.
Response 2: Thank you for raising this question. Based on the analysis of relevant work, we categorize the methods according to their underlying approaches, dividing them into four main parts:
- Utilizing attention mechanisms as the primary means to enhance feature extraction and multi-scale detection capabilities.
- Combining attention mechanisms with other techniques to reduce parameters while improving feature extraction.
- Enhancing training data to minimize model overfitting.
- Improvements to convolutional layers, backbones, and loss functions are made to boost the model's detection accuracy.
Page 2 at line 94 to page 4 at line 195:
Ship detection is a specialized area within object detection that requires balancing two key objectives: accuracy and real-time performance. Given the significant variation in ship sizes, the network must also have strong multi-scale detection capabilities. One of the most effective ways to enhance a neural network's object detection accuracy is by improving its feature extraction capabilities, often achieved by increasing the network's depth. However, this approach also increases computational complexity and the number of parameters, which can negatively impact the network's real-time performance. Consequently, numerous scholars have conducted extensive research on improving accuracy, real-time efficiency, and multi-scale detection capabilities of networks.
2.1 Attention mechanism in ship detection
The attention mechanism plays a crucial role in enhancing feature extraction and multi-scale detection capabilities. For instance, Shen et al[14]. introduced a Multiple Information Perception-based Attention Module (MIPAM). Their approach incorporates channel-level information collection through global covariance pooling and channel-wise global average pooling, while spatial-level information is collected in a similar way. This method enriches feature representation, leading to improved detection accuracy when integrated into the YOLO detector. Due to the diverse shapes of ships and the complexities of environmental interference, multi-scale detection has become an essential capability for ship detection networks. Many experts utilize attention mechanisms to dynamically adjust the weights within a network, designing and implementing them to enhance the network's ability to capture important information across different scales. In [15], the author proposed an Attention Feature Filter Module (AFFM), which uses attention supervision generated from high-level semantic features in the feature pyramid to highlight information-rich targets, forming a spatial attention mechanism. Unlike traditional attention mechanisms such as CBAM, the attention signals in AFFM are derived from higher-level feature maps, which better represent the distinctive characteristics of nearshore ships. Guo et al[16]. utilized sub-pixel convolution, sparse self-attention mechanisms, channel attention, and spatial attention mechanisms to enhance semantic features layer by layer, which ensures that feature map contains richer high-level and low-level semantic information, effectively improving the detection performance of small ships. Yao et al[17]. designed a feature enhancement module based on channel attention, increasing the emphasis on ship features and expanding the receptive field through an adaptive fusion strategy. This enhances spatial perception for ships of varying sizes. Li et al[18]. developed an adaptive spatial channel attention module, effectively reducing the interference of dynamic background noise on large ships. Additionally, they designed a boundary-box regression module with gradient thinning, improving gradient sensitivity and multi-scale detection accuracy. Li et al[19]. took a more intuitive approach by incorporating the Transformer into the YOLOv5 backbone to enhance feature extraction. Zheng et al[20]. integrated a local attention module into SSD to improve the detection accuracy of smaller models. Li et al[21]. enhanced YOLOv3 by adding the CBAM attention mechanism to the backbone, enabling the model to focus more on the target and thereby improving detection accuracy.
2.2 Attention mechanism combined with other improvements
Some authors combine attention mechanisms with other techniques, such as improved loss functions and enhanced convolution methods, to not only boost feature extraction capabilities but also reduce network parameters and improve real-time performance. For example, Zhao et al[22]. introduced the SA attention mechanism into YOLOv5n to enhance feature extraction and replaced standard convolution in the neck with Ghost Conv, reducing both network complexity and computational load. In [23], the author proposed a lightweight LWBackbone to decrease the number of model parameters and introduced a hybrid domain attention mechanism, which effectively suppresses complex land background interference and highlights target areas, achieving a balance between precision and speed. Ye et al[24]. incorporated the CBAM attention module into the backbone of YOLOv4, replaced CIOU with EIOU, and substituted non-maximum suppression (NMS) with soft-NMS to reduce missed detections of overlapping ships. Bowen Xing et al[25]. added the CBAM attention mechanism to FasterNet, replaced the YOLOv8 backbone, and introduced lightweight GSConv convolution in the neck to enhance feature extraction and fusion. Additionally, they improved the loss function based on ship characteristics and integrated MPDIoU into the network, making it more suitable for ship detection.
2.3 Data enhancement prevent overfitting
Some experts and scholars focus on ship data itself, utilizing data enhancement and preprocessing as primary methods to improve network detection performance, along with other enhancements. For instance, Zhang et al[26]. developed a new data enhancement algorithm called Sparse Target Mosaic, which improves training samples. By incorporating a feature fusion module based on attention mechanisms and refining the loss function, they were able to enhance detection accuracy. Gao et al[27]. applied gamma transform to preprocess infrared images, increasing the gray contrast between the target and background. They also replaced the YOLOv5 backbone with MobileNetV3, reducing parameters by 83% without significantly compromising detection performance. Chen et al[28]. designed pixel-space data enhancement in a two-stage target detection network, using set transformation and pixel transformation to enhance data diversity and reduce model overfitting. This approach improved network focus and accuracy, achieving a remarkable mAP of 99.63%. Qiu et al[29]. addressed the singleness of the dataset’s image style in ship detection datasets by proposing an anti-attention module. They input the original feature layer into a trained convolutional neural network, filtered the output weights, and removed feature layers that negatively impacted detection. This led to improvements in both mAP and F1-score.
2.4 Improvement in lightweight
Some scholars focused on improving convolutional methods, backbone, and loss functions to enhance ship detection performance. For example, Li et al[30]. integrated OD-Conv into the YOLOv7 backbone, effectively addressing the issue of complex background interference in ship images, thereby improving the model's feature extraction capabilities. Additionally, they introduced the space-to-depth structure in the head network to tackle the challenge of detecting small and medium-sized ship targets. These improvements led to a 2.3% increase in mAP compared to the baseline. Zheng et al[31]. proposed a differential-evolution-based K-means clustering method to generate anchors tailored to ship sizes. They also enhanced the loss function by incorporating Focal Loss and EIOU, resulting in a 7.1% improvement in average precision compared to YOLOv5s. Shi et al[32]. introduced the theta-EIOU loss function, which enhances the network's learning and representation capabilities by reconstructing the bounding box regression loss function, improving background partitioning, and refining sample partitioning. The improved method outperformed the original YOLOX network. Zhang et al[33]. incorporated a multi-scale residual module into YOLOv7-Tiny and designed a lightweight feature extraction module. This reduced the number of parameters and computational load of the backbone while improving feature extraction accuracy. They used Mish and SiLU activation functions in the feature extraction module to enhance network performance, and introduced CoordConv in the neck network to reduce feature loss and more accurately capture spatial information. Zheng et al[34]. replaced YOLOv5s' original feature extraction backbone with the lightweight MobileNetV3-Small network and reduced the depth-separable convolutional channels in the C3 module to create a more efficient feature fusion module. As a result, the final network had 6.98MB fewer parameters and an improved mAP compared to YOLOv5s.
Comments 3: The attention mechanism section in the fourth chapter lacks emphasis on key points. The title of section 4.2 should be "Attention DLKA Module," with a detailed explanation of this module, while the discussion on deformable convolution could be reduced.
Response 3: Thank you for pointing out this error. The title of section 4.2 has been updated. deformable convolution, now part of the module, is integrated into the module for explanation, and additional details about the module itself have been included.
Page 8 at line 281 to 289:
In the task of ship target detection, the size and shape of different ship types can vary significantly, encompassing a wide range of morphologies and structures. This diversity necessitates that detector be capable of adapting to multiple forms, which increases model complexity and makes it challenging to generalize across different ship types. Additionally, the maritime environment, including oceans, harbors, and other water contexts, is often complex and dynamic, with potential disturbances such as waves, other vessels, and infrastructure. Moreover, the diversity in ship sizes and shapes can lead to sample imbalance in the training data, resulting in suboptimal model performance for certain ship categories.
Page 9 at line 299 to 308:
The structure of split-DLKA is illustrated in the figure. A tensor x of size [B,C,H,W] is divided into i subsets along the channel dimension, denoted xi, i=1,2,... where i=2 . Here, each subset xi has C/2 input channels, which are passed separately to the deformable DW module.
To expand the receptive field of the neural network, large convolutional kernels of size 5x5 and 7x7 are applied to deformable DW module, which can adaptively adjust its sampling positions to better align with the local features of the input data, thereby enhancing feature extraction capability. This improvement allows the model to adapt more effectively to geometric transformations, enabling the network to better handle irregularly shaped objects or features.
Page 10 at line 329:

Figure 4. Structure of split-DLKA
Comments 4: The section on the loss function in the fourth chapter should include diagrams to improve readability and help readers understand the concepts better.
Response 4: Thank you for your questions. We have added two diagrams: one illustrating the detection box and ground truth, and the other showing the curves of outlier degree and gradient gain under different hyperparameters.
Page 11 at line 343:
Figure 5. Ground truth, predict box and their minimum enclosing rectangle.
Page 12 at line 363:

Figure 6. Curves of outlier degree and gradient gain under different hyperparameters.
Comments 5: The selection of hyperparameters for the loss function in the fourth chapter is missing comparative analysis experiments.
Response 5: Thank you for your query. We have added comparative experiments on hyperparameter selection and provided explanations for the rationale behind the chosen hyperparameters.
Page 11 at line 375 to 362:The WIOUv3 loss function replaces the original loss function, thereby reducing the model's overfitting and enhancing its detection performance. Additionally, assigning a small gradient gain to the anchor box with a large outlier degree will effectively prevent large harmful gradients from low-quality examples. It can be seen from Figure 2 that when α=1.9,δ=3 , we can get the biggest gradient and the smallest outlier degree. The impact of hyperparameters on experimental results is shown in Table 1. As can be seen from the table, when α=1.9,δ=3 the model has a highest detection accuracy.
Page 21 at line 532:
Table 10. Effects of different hyperparameters on model performance.
|
α |
δ |
Precision |
Recall |
mAP@0.5 |
mAP@0.5:0.95 |
|
2.5 |
2 |
0.84 |
0.82 |
0.891 |
0.539 |
|
1.9 |
3 |
0.85 |
0.85 |
0.912 |
0.560 |
|
1.6 |
4 |
0.83 |
0.85 |
0.901 |
0.555 |
|
1.4 |
5 |
0.83 |
0.84 |
0.894 |
0.545 |
Comments 6: In Table 3, which compares the algorithm's performance with YOLOv5, the algorithm has fewer parameters and less computational load than YOLOv5, yet it has a lower FPS, and this discrepancy should be explained.
Response 6: Thank you for pointing out this issue. According to the statistics of the average inference time on the test set, our method increases the inference time by 0.5 ms compared to YOLOv5. This increase is due to the use of larger convolution kernels (5x5 and 7x7) in split-DLKA, which results in longer computation times. While our method sacrifices some inference time, it achieves an improvement in detection accuracy. The following analysis of the inference time for both methods has been added to the paper:
Page 15 at line 432 to 438: The FPS of our method is 20 lower than that of YOLOv5s. As shown in Table 4, the main increase in time occurs during the prediction frame generation stage, due to the use of large convolution kernels (5x5 and 7x7) in the split-DLKA module, which leads to longer inference times. We traded off some inference speed to achieve higher detection accuracy.
Table 4. Analysis of inference time between YOLOv5s and our method.
|
Method |
Preprocess |
Inference |
NMS |
|
YOLOv5s |
0.3 ms |
3.7 ms |
0.7 ms |
|
ours |
0.3 ms |
4.1 ms |
0.8 ms |
Preprocess means affine transformation, inference means generating prediction boxes and NMS is used to remove redundant prediction boxes.
Comments 7: The evaluation metrics should include the final model's weight size to highlight the ease of deployment on edge devices.
Response 7: Thank you for raising this question. As you mentioned, the size of the model's weights should be considered as part of the evaluation metrics. A smaller model size indicates lower computing power requirements. Therefore, I included weight size as an evaluation metric in the comparative experiment of different methods and provided an explanation for it.
Page 18 at line 474 to 479:
For example, YOLOv9c has 12 times more parameters, 15.9 times more FLOPs and 11.7 times more weight size to the proposed method. Despite these differences, the proposed method achieves competitive performance, trailing YOLOv9c by 7% in terms of mAP@0.5. Generally, a larger weight size implies that more memory and computational power are required to run the network, which is a significant drawback for unmanned surface vessels.
Page 17 at line 465:
Table 5. Contrast experiments.
|
Model |
Parameters |
FLOPs |
Precision |
Weight size |
Recall |
F1 |
mAP@0.5 |
mAP@0.5:0.95 |
FPS |
|
YOLOv5m |
20.89M |
48.3G |
0.879 |
42.2MB |
0.88 |
0.879 |
0.944 |
0.602 |
119 |
|
YOLOv5l |
46.17M |
108.3G |
0.889 |
92.8MB |
0.89 |
0.889 |
0.946 |
0.609 |
66 |
|
yolov4-tiny |
5.89M |
6.8G |
0.878 |
24.3MB |
0.45 |
0.595 |
0.689 |
0.303 |
370 |
|
Nanodet-plus |
0.95M |
1.2G |
0.667 |
2.44MB |
0.62 |
0.642 |
0.641 |
0.341 |
55 |
|
YOLOv5s-efficientnetv2 |
5.6M |
5.6G |
0.738 |
11.5MB |
0.74 |
0.738 |
0.809 |
0.431 |
208 |
|
YOLOv9c |
25M |
103.7G |
0.98 |
51.8MB |
0.97 |
0.974 |
0.99 |
0.82 |
58 |
|
ours |
2.02M |
6.5G |
0.89 |
4.4MB |
0.84 |
0.864 |
0.920 |
0.563 |
192 |
Response to Comments on the Quality of English Language
Point 1: Moderate editing of English language required.
Response 1: Thanks for your suggestion, we have revised the manuscript according to the polishing comments of professional institutions. The following is the proof of polishing.

Round 2
Reviewer 2 Report
Comments and Suggestions for Authors
Accepted after revisions for grammar and spelling
Comments on the Quality of English LanguageRevised some grammar and spelling